# EvoEngineer: Mastering Automated CUDA Kernel Code Evolution with Large Language Models

## Abstract

CUDA kernel optimization has become a critical bottleneck for AI performance, as deep learning training and inference efficiency directly depends on highly optimized GPU kernels. Despite the promise of Large Language Models (LLMs) for automating kernel optimization, this field suffers from a fragmented ecosystem of isolated and incomparable approaches with unclear problem formulations. Furthermore, general-purpose LLM code evolution methods cannot meet strict correctness requirements of CUDA kernel optimization. We address these fundamental challenges by first formalizing CUDA kernel optimization as a code optimization task with a clear objective, constraints, and evaluation metrics. We then establish the first systematic LLM-based code evolution framework, EvoEngineer, that provides guidance for designing and adapting optimization strategies to achieve a balance between performance and correctness. Finally, we implement a kernel optimization system based on this framework and conduct extensive experiments on 91 real-world CUDA kernels. Our results demonstrate that EvoEngineer achieves a principled balance between performance and correctness, with the highest averaged median speedup of $2.72\times$ over baseline CUDA kernels and a code validity rate of **69.8%**, outperforming existing methods on both dimensions. Our method achieves a maximum speedup of $36.75\times$ among all operations over PyTorch kernels and delivers the highest speedup on **25 (59.5%)** of 42 operations that achieve over $2\times$ acceleration. Our code is available at `https://anonymous.4open.science/r/EvoEngineer_open-FD4E/`.

## 1 Introduction

CUDA kernel performance has become the critical bottleneck constraining the efficiency of AI training and inference. As foundation models continue scaling to unprecedented sizes (Guo et al., 2025; Jaech et al., 2024), computational demands necessitate maximum GPU utilization efficiency, where even marginal improvements in kernel performance can yield substantial reductions in computational costs. However, manual kernel optimization requires deep expertise across GPU architectures, memory hierarchies, parallelization patterns, and hardware-specific features (Navarro et al., 2020; Hennessy & Patterson, 2011), constituting a major obstacle to scaling AI systems.

The kernel code optimization landscape presents extreme complexity, involving intricate trade-offs between memory coalescing, thread divergence, occupancy optimization, and register usage (Ujaldón, 2016; Huang et al., 2021; Zhao et al., 2022). Furthermore, the rapid evolution of GPU architectures intensifies this challenge, requiring optimization strategies to be continuously adapted for the distinct characteristics of each new hardware (Chen et al., 2018). This complexity calls for sophisticated automated approaches capable of navigating the discrete, structured, and semantically-sensitive nature of kernel code space.

Large language models (LLMs) have demonstrated remarkable capabilities in code generation, offering promise for addressing this automation challenge. Recent kernel-specific approaches have begun integrating LLMs with iterative search techniques (Ouyang et al., 2025; Lange et al., 2025; Chen et al., 2025a;b; Li et al., 2025; Andrews & Witteveen, 2025; Wang et al., 2025), achieving encouraging results in performance improvement. Meanwhile, general-purpose code evolution methods

such as Evolution of Heuristics (EoH) (Liu et al., 2024), FunSearch (Romera-Paredes et al., 2024), and related evolutionary approaches (Yao et al., 2024; Novikov et al., 2025; Guo et al., 2024b) have achieved success in domains like mathematical problem solving and heuristic algorithm design and have begun exploring kernel optimization (Novikov et al., 2025).

However, these developments reveal a fundamental limitation: the lack of systematic frameworks for understanding the effectiveness of different optimization strategies. Kernel-specific methods suffer from tightly coupled evaluation processes and unclear problem formulations that prevent systematic analysis and fair comparisons. Meanwhile, general-purpose code evolution methods have been confined to problems with loose correctness requirements and provide no principled guidance for strategy selection and design.

This absence of systematic frameworks creates a critical gap: *how can we systematically select appropriate optimization strategies for different problem contexts to reliably improve both performance and correctness of automated kernel code generation?*

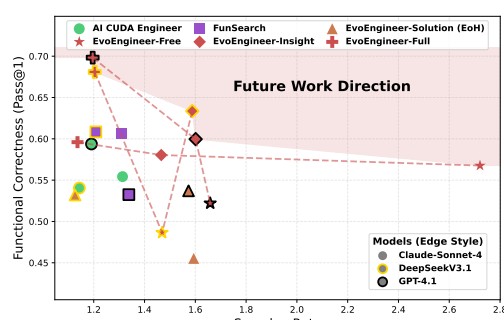

Figure 1: Trade-off between speedup and functional correctness across different optimization strategies, showing the dominance of EvoEngineer variants.

To address these challenges, we propose the first systematic framework, EvoEngineer, for analyzing and selecting optimization strategies in LLM-based code evolution. Our key insight is to decompose code evolution into two orthogonal components: a two-layer traverse technique and population management. This decomposition enables independent analysis of each component. Moreover, the two-layer design of the traverse technique clearly separates optimization strategy from prompt engineering, helping us navigate the code space effectively.

We achieve substantial improvements using three strategy instances under EvoEngineer's guidance across 91 real-world CUDA kernel operations spanning six distinct categories. The experimental results demonstrate that systematic optimization strategy selection leads to significant performance gains and correctness improvements, validating our framework's ability to guide systematic optimization strategy design, as illustrated in Figure 1.

Our work advances the field through three key contributions:

- **Systematic Framework:** The first comprehensive framework, EvoEngineer, for analyzing and selecting optimization strategies in LLM-based code evolution, decomposing methods into traverse techniques and population management while providing clear criteria for balancing performance and correctness.

- **Open-Source Platform:** A modular system implementing our systematic approach and kernel optimization task that enables reproducible research and fair comparison of different code evolution techniques in CUDA kernel code optimization.

- **Empirical Validation:** Large-scale validation across 91 real-world CUDA kernels spanning six categories, demonstrating that systematic strategy selection achieves substantial performance improvements while establishing practical guidelines for optimization method selection in real-world applications.

## 2 RELATED WORK

**CUDA Kernel Optimization Challenges.** The kernel is a unit of execution in heterogeneous parallel computing, defined as a parallel function that runs on a specialized processing unit like a GPU (Chow & Kohler, 1979). Achieving optimal kernel performance requires deep expertise across multiple complex dimensions: managing memory hierarchies, scheduling thread blocks efficiently, leveraging hardware-specific instruction sets and features like NVIDIA's Tensor Cores(Navarro et al., 2020; Hennessy & Patterson, 2011; Ujaldón, 2016).

## 2.1 LLM-BASED CODE OPTIMIZATION APPROACHES

**Kernel-Specific Optimization Methods.** Recent research has turned to LLMs for automated kernel optimization, leveraging their powerful code generation capabilities to automate high-performance optimization. Current kernel-specific approaches include evaluation benchmarks like Kernel-Bench(Ouyang et al., 2025) and iterative optimization methods such as AI CUDA Engineer(Lange et al., 2025), NVIDIA's closed-loop workflows(Chen et al., 2025a), and various domain-specific systems(Chen et al., 2025b; Li et al., 2025; Andrews & Witteveen, 2025; Wang et al., 2025) (see Appendix A.1 for detailed descriptions). These methods typically employ iterative search strategies where LLMs generate kernel variants based on performance feedback and compilation results.

**General-Purpose Code Evolution Methods.** Developments in general-purpose code evolution have demonstrated the potential of integrating LLMs into systematic evolutionary search frameworks. Methods such as Evolution of Heuristics (EoH)(Liu et al., 2024), FunSearch(Romera-Paredes et al., 2024), and related evolutionary approaches(Yao et al., 2024; Novikov et al., 2025; Guo et al., 2024b) have achieved notable successes in domains like mathematical problem solving and heuristic algorithm design. These approaches leverage iterative refinement and population-based search to evolve code solutions, with some recent work beginning to explore kernel optimization applications(Novikov et al., 2025).

**Limitations of Current Approaches.** Despite promising results, both kernel-specific and general-purpose approaches share critical limitations. Kernel-specific methods suffer from tightly coupled evaluation-optimization process that prevent analysis of optimization strategy effectiveness, while general-purpose code evolution methods face significant scalability challenges when applied to domains demanding strict correctness and performance requirements.

In general, they both lack systematic frameworks for understanding which optimization strategies work effectively in different problem contexts, preventing principled strategy selection to improve both performance and correctness of automated kernel code generation.

# 3 AUTOMATED CUDA KERNEL CODE OPTIMIZATION

## 3.1 PROBLEM FORMULATION

**Code Optimization.** Code optimization involves searching through the space of possible program implementations to optimize specific objectives while maintaining correctness. We formalize this as a constrained optimization problem:

$$p^* = \arg \min_{p \in \mathcal{S}} f(p)$$
$$\text{s.t.} \quad g(p) = 0 \tag{1}$$

where $p$ represents a candidate program, $\mathcal{S}$ is the space of all possible code implementations, $f(p)$ measures performance objectives, and $g(p) = 0$ encodes correctness and feasibility constraints.

The constraint function $g(p)$ encompasses multiple critical requirements:

- **Syntactic Validity:** Code must parse and compile without errors
- **Functional Correctness:** Output must match reference implementation across test cases

This formulation reveals a fundamental challenge: the vast majority of $\mathcal{S}$ consists of invalid solutions, making effective navigation strategies crucial for success.

**Search Space $\mathcal{S}$.** Traditional approaches operate in transformed representation spaces: continuous embedding spaces for gradient-based optimization (Allamanis et al., 2018), or discrete AST/syntax tree spaces for genetic programming (Keller & Banzhaf, 1999; Koza, 1994).

LLM-based approaches enable direct search in the raw text space $\mathcal{S}_{\text{text}}$, where each program $p$ is simply a string of code. This transformation has spawned numerous code evolution methods (Liu et al., 2024; Romera-Paredes et al., 2024; Yao et al., 2024; Novikov et al., 2025) that can perform direct modifications without intermediate representations.

Table 1: Comparative analysis of LLM-based methods for kernel optimization. ✔ indicates presence, ✗ indicates absence.

| Category | Method | | M1 | M2 | M3 | M4 | | Token Usage | Application Area |
|---|---|---|---|---|---|---|---|---|---|
| **Traditional** | Genetic Programming (Koza, 1994) | ‖ | ✗ | ✔ | ✔ | ✔ | ‖ | N/A | General Code Generation |
| **Generative AI** | AlphaCode (Li et al., 2022) | ‖ | ✗/✔[‡] | ✗ | ✗ | ✗ | ‖ | N/A | General Code Generation |
| **LLM-Based** | AI CUDA Engineer (Lange et al., 2025) | ‖ | ✔ | ✔ | ✗ | ✗ | ‖ | High | CUDA Kernel Optimization |
| | EoH (Liu et al., 2024) | ‖ | ✔ | ✔ | ✔ | ✗ | ‖ | Low | Heuristic Algorithm Design |
| | FunSearch[*] (Romera-Paredes et al., 2024) | ‖ | ✔ | ✔ | ✔ | ✗ | ‖ | Low | Mathematical Function Search |
| | **EvoEngineer (Ours)** | ‖ | ✔ | ✔ | ✔ | ✔ | ‖ | Configurable | General Code Generation |

**M1 (Text Search Space):** Search Space indicates whether the method operates directly in text space.
**M2 (Iterative Search):** Whether the method employs iterative improvement over multiple generations.
**M3 (Open Source):** Availability of code and evaluation framework for reproducibility and extension.
**M4 (Systematic Framework):** Whether has systematic framework for guidance of selection of optimization strategies
[‡]: AlphaCode uses text space for generation but relies on operating in other representation space.
[*]: Also the core technique behind AlphaEvolve Novikov et al. (2025).

## 3.2 RESEARCH GAP ANALYSIS

**Research Gap.** To understand the state of LLM-based kernel code optimization, we analyze existing approaches across four key dimensions: search space utilization, iterative capability, reproducibility, and systematic strategy guidance in Table 1. Existing methods lack a systematic framework for principled guidance on optimization strategy (**M4**). This can lead to two serious consequences: *resource inefficiency* and *strategy blindness*.

*Resource Inefficiency.* Methods like AI CUDA Engineer (Lange et al., 2025) employ complex prompt engineering while lacking understanding of optimization strategies, leading to unnecessary token usage without corresponding speedup gains (See Figure 4 in Section 5.3).

*Strategy Blindness.* Existing methods often rely on ad-hoc combinations of search techniques without principled guidance on their application. This prevents the design of strategies tailored to specific problem contexts and impedes advancement of LLM-based code evolution.

## 4 EVOENGINEER: A SYSTEMATIC FRAMEWORK FOR LLM-BASED CODE EVOLUTION

We present EvoEngineer, a systematic framework that decomposes LLM-based code evolution into two orthogonal components: *traverse techniques* for navigation strategies and *population management* for solution maintenance. An overview of the framework is shown in Figure 2.

Our methodology is structured in three parts. First, we establish our systematic framework with traverse techniques and population management, introducing a novel two-layer design for traverse techniques (Section 4.1). Second, we analyze existing methods through this framework and configure three key EvoEngineer strategies that embody different strategic choices (Section 4.2). Finally, we present our modular system implementation that enables reproducible research and fair comparison of code evolution techniques in kernel optimization (Section 4.3).

### 4.1 FRAMEWORK DESIGN

Aligned with the evolutionary algorithm paradigm, our framework decomposes any code evolution method into two fundamental components: *traverse techniques* that navigate the discrete, structured code space $\mathcal{S}_{\text{text}}$ and *population management* strategies that maintain and select candidate solutions across generations.

### 4.1.1 TRAVERSE TECHNIQUES: A TWO-LAYER DESIGN

**Motivation for Layered Design.** Existing LLM-based code evolution methods (Liu et al., 2024; Guo et al., 2024a; Yao et al., 2024) often conflate optimization strategy with prompt engineering. The operators in these methods often lack clear definitions, in contrast to traditional optimization domains where search operators are well-defined.

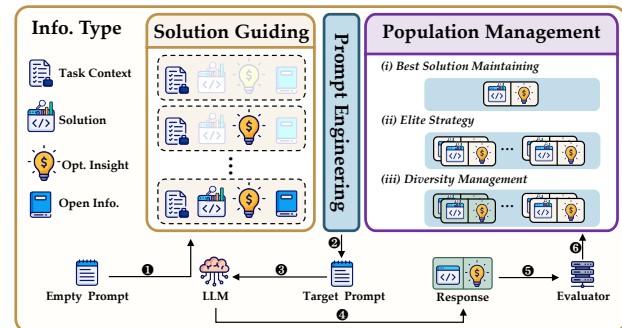

Figure 2: **EvoEngineer framework for LLM-based code evolution.** The framework includes two orthogonal components: *traverse techniques* for navigation and *population management* for solution maintenance. Traverse techniques consist of a *solution guiding layer* that defines navigation strategy and a *prompt engineering layer* that communicates this strategy to the LLM.

To address this, we decompose traverse techniques into two distinct layers: *Solution Guiding Layer* that defines the high-level strategy for navigating code space, and *Prompt Engineering Layer* that specifies how to communicate this strategy to the LLM through prompt design.

**Solution Guiding Layer:** The solution guiding layer determines what information to include in prompts to effectively guide the LLM towards promising regions of $\mathcal{S}_{\text{text}}$. We categorize this information into two types: *closed-world information* that is self-contained during the search process and *open-world information* that can be flexibly provided externally.

*Closed-world Information.* We identify three key types of closed-world information that can guide LLMs in code evolution: *1)* the current task context encoding specific optimization goals and constraints, *2)* historical high-quality solutions recording previous trials, and *3)* optimization insights containing design rationales and LLM reasoning processes.

*Open-world Information.* Open-world information refers to additional context such as domain knowledge that can be provided through retrieval-augmented methods (Lewis et al., 2020; Karpukhin et al., 2020) or human experts. In this paper, we focus on closed-world information while leaving open-world information for future work.

**Prompt Engineering Layer:** The prompt engineering layer translates the high-level strategy from the solution guiding layer into concrete prompts for the LLM. This involves decisions on prompt structure, content, and formatting to effectively communicate the intended guidance. We follow common practices in prompt engineering (Liu et al., 2023) such as explicit instructions to enhance LLM performance and do not provide further implementation details in this paper.

**Operator Design vs. Two-Layer Traverse Techniques.** We distinguish our two-layer framework from traditional operator design approaches used in existing LLM-based code evolution methods (Liu et al., 2024; Romera-Paredes et al., 2024). Traditional operator design focuses on mimicking evolutionary operators (crossover, mutation) in the LLM context. However, this approach suffers from two fundamental issues: *(1) Strategy-Implementation Conflation:* Operators mix high-level optimization strategy with low-level prompt engineering, making it difficult to analyze what drives effectiveness. *(2) Unvalidated Assumptions:* No empirical evidence demonstrates that LLMs can meaningfully perform crossover or mutation operations as traditionally defined.

Our framework addresses these limitations by: *(1)* clearly separating optimization strategy from implementation details, *(2)* providing systematic methods for organizing and utilizing information, and *(3)* enabling sophisticated prompt engineering while maintaining clear strategy definitions.

### 4.1.2 POPULATION MANAGEMENT

Population management defines how candidate solutions are maintained, selected, and evolved across generations. Unlike traverse techniques that focus on generating new solutions, population management determines which solutions to preserve.

Table 2: Framework-based analysis of existing methods. ✔ indicates presence, ✘ indicates absence.

| Method | Solution Guiding Layer | | | |
| --- | --- | --- | --- | --- |
| | Closed-world | | | Open |
| | I1 | I2 | I3 | I4 |
| AI CUDA Engineer (Lange et al., 2025) | ✔ | $> 5$ | ✘[*] | Inter-op. |
| EoH (Liu et al., 2024) | ✔ | $2 - 3$ | ✘[*] | ✘ |
| FunSearch (Romera-Paredes et al., 2024) | ✔ | 2 | ✘ | ✘ |

I1: Task Context; I2: Historical Solutions; I3: Optimization Insights; I4: Domain Knowledge
[*]: Generate insights but don't leverage them explicitly

Table 3: EvoEngineer configurations on closed-world information usage.

| Configuration | Info. Types | | |
| --- | --- | --- | --- |
| | I1 | I2 | I3 |
| EvoEngineer-Free [1] | ✔ | ✘ | ✘ |
| EvoEngineer-Insight [1] | ✔ | ✘ | ✔ |
| EvoEngineer-Solution (EoH) [2] | ✔ | ✔ | ✘ |
| EvoEngineer-Full [2] | ✔ | ✔ | ✔ |

I1: Task Context; I2: Historical Solutions; I3: Optimization Insights
[1]: Best solution maintaining; [2]: Elite preservation strategy

We categorize population management strategies within current methods: *(1) single-solution strategy* that maintains only the current best solution, *(2) elite preservation strategy* that keeps a small set of high-performing solutions, *(3) diversity maintenance strategy* that keeps diver sets of solutions to explore different regions of the search space.

## 4.2 FRAMEWORK-BASED ANALYSIS AND NEW CONFIGURATIONS

We analyze existing LLM-based code evolution methods through our framework and systematically configure three key EvoEngineer strategies to incorporate different information in traverse techniques and population management.

**Solution Guiding Layer Analysis.** As the solution guiding layer is the most critical component of traverse techniques, we focus our analysis on the information utilized by each method. Table 2 summarizes our analysis results.

All methods incorporate basic task context (**I1**), demonstrating universal recognition of this fundamental requirement. Methods vary dramatically in their utilization of historical solutions (**I2**). AI CUDA Engineer leverages the largest set ($> 5$ solutions), EoH maintains moderate usage ($2 - 3$ solutions), while FunSearch uses minimal historical information (2 solutions). For optimization insights (**I3**), although AI CUDA Engineer and EoH both require the LLM to produce solution-insight pairs, neither method explicitly leverages these insights to guide the search process. For open-world information (**I4**), only AI CUDA Engineer incorporates inter-operational knowledge, while both EoH and FunSearch operate without external domain-specific information.

**Framework-Based Configurations.** Based on our framework analysis, we configure three key configurations of optimization strategies. Our configurations systematically explore different information types, as summarized in Table 3. We configure three EvoEngineer variants: *(1) EvoEngineer-Free* utilizes only task context (**I1**) with low-complexity prompting and best-solution maintenance, prioritizing exploration over correctness; *(2) EvoEngineer-Insight* leverages optimization insights (**I3**) extracted as separate information sources rather than solution-bound pairs, using single best-solution maintenance; and *(3) EvoEngineer-Full* integrates both historical solutions (**I2**) and optimization insights (**I3**) with elite preservation strategy, expected to achieve the highest correctness through comprehensive information integration.

## 4.3 A MODULAR KERNEL OPTIMIZATION SYSTEM IMPLEMENTATION

To validate our theoretical framework and advance systematic CUDA kernel optimization research, we have implemented a modular kernel optimization system. The system architecture emphasizes modularity and extensibility, allowing researchers to evaluate different optimization strategies.

**Modular Architecture Design.** As illustrated in Figure 3, our system follows three-step workflow: *1) Task Configuration*, where critical task details such as GPU types, CUDA versions, and evaluation metrics are specified for the kernel optimization task; *2) Solution Generation*, where we implement the traverse techniques and population management strategies defined in our EvoEngineer framework; and *3) Solution Evaluation*, where generated kernels are compiled, executed, and evaluated for performance and correctness.

Figure 3: **Overview of our modular kernel optimization system.** This system follows a three-step workflow: *(1) Task Configuration*, *(2) Solution Generation*, and *(3) Feedback Collection*.

Specifically, we employ a two-stage evaluation process: *1) Compilation Check* to ensure syntactic validity by compiling the code, and *2) Functional Testing* to verify correctness by running five test cases and comparing outputs against reference PyTorch implementations. Finally, performance metrics (*e.g.*, execution time in this paper) are collected for valid and correct solutions over 100 runs to take the average to ensure statistical significance.

## 5 EXPERIMENTAL RESULTS

### 5.1 EXPERIMENT SETUP

This section details the experimental configuration, including hardware specifications, software environments, dataset characteristics, baseline methods, parameter settings, and evaluation metrics.

**Experimental Environment.** All experiments were conducted on a dedicated system with AMD EPYC 7542 32-Core Processor and NVIDIA RTX 4090 GPU, running Ubuntu 22.04 LTS with Python 3.11, PyTorch 2.4.0, and CUDA 12.4.1. Complete hardware specifications, software dependencies, and reproducibility guidelines are provided in Appendix A.2.

**Dataset.** We crafted a dataset containing 91 distinct deep learning-related operations derived from the KernelBench benchmark (Ouyang et al., 2025). For each operation, our prepared dataset includes a reference Python implementation for correctness verification and a corresponding initial C++/CUDA implementation to serve as the starting point for optimization (associated with our NVIDIA RTX 4090 GPU). The operations are categorized into 6 groups by computational complexity and functionality, as shown in Table 5 in Appendix A.3.

**Methods.** We compare EvoEngineer against state-of-the-art kernel-specific optimization methods and general-purpose code evolution methods. For kernel-specific methods, we include AI CUDA Engineer (Lange et al., 2025), as there is no implementation for it available, we re-implemented it based on the description in the original paper, with replication details provided in Appendix A.8. For general-purpose methods, we include the famous FunSearch (Romera-Paredes et al., 2024) and EoH (Liu et al., 2024), representing different paradigms of automated code evolution methods.

**Parameter Setting.** To ensure a fair comparison, we constrain the maximum number of optimization trials to 45 for each kernel across all methods. For the Large Language Model, we incorporate GPT-4.1 from OpenAI, DeepSeek-V3.1 from DeepSeek, and Claude–Sonnet-4 from Anthropic. The detailed parameter settings and configurations for each method are provided in Appendix A.4.

**Evaluation Metric.** The primary performance metric is the median speedup across different kernels. Using median mitigates the influence of outliers, providing a robust measure of performance improvement. When a method fails to produce a kernel that outperforms the baseline, we assign a speedup of 1.0 for that kernel, ensuring that failures do not skew the overall performance assessment. The averaged speedup is computed over three independent runs to account for variability in performance measurements.

Table 4: **Overall Results of different methods using different LLMs.** Speedup performance is measured by *Speedup Count* and *Median Speedup Rate*. Code validity performance is assessed through *Compilation Success* and *Functional Correctness*. All results are averaged across three independent runs. The best results within each metric and model are highlighted in **bold** with gray background ▨; the second-best results are underlined.

| Speedup Analysis | | Speedup Count | | | | | | | Median Speedup Rate | | | | | | |
| --- | --- | --- | --- | --- | --- | --- | --- | --- | --- | --- | --- | --- | --- | --- | --- |
| | | 1 | 2 | 3 | 4 | 5 | 6 | Overall | 1 | 2 | 3 | 4 | 5 | 6 | Overall |
| **GPT-4.1** | AI CUDA Engineer | 16.7 | 24.0 | 19.7 | 13.7 | 7.0 | 3.0 | 84.0 | 1.83 | 1.13 | 1.11 | 4.02 | 1.25 | 21.34 | 1.19 |
| | FunSearch | 14.0 | 21.3 | 19.0 | 13.0 | 7.0 | 3.3 | 77.7 | 4.30 | 1.14 | 2.69 | 1.14 | | 32.47 | 1.34 |
| | EvoEngineer-Solution (EoH) | 16.7 | 22.0 | 20.7 | 13.7 | 6.0 | 3.3 | 82.3 | 2.82 | 1.35 | 1.41 | 2.22 | 1.70 | 14.64 | 1.57 |
| | EvoEngineer-Free | 13.7 | 22.0 | 20.7 | 14.3 | 5.7 | 4.0 | 80.3 | 4.75 | 1.42 | 1.29 | 3.22 | 4.63 | 38.65 | 1.66 |
| | EvoEngineer-Insight | 13.7 | 18.0 | 20.7 | 13.0 | 7.0 | 3.0 | 76.7 | 3.41 | 1.31 | 1.47 | 2.57 | 1.41 | 27.86 | 1.60 |
| | EvoEngineer-Full | 15.0 | 18.7 | 20.0 | 13.7 | 7.0 | 3.0 | 77.3 | 2.41 | 1.07 | 1.11 | 2.12 | 1.24 | 13.66 | 1.20 |
| **DeepSeekV3.1** | AI CUDA Engineer | 15.3 | 22.3 | 18.3 | 13.3 | 6.3 | 2.3 | 78.0 | 1.51 | 1.08 | 1.09 | 2.07 | 1.29 | 1.10 | 1.14 |
| | FunSearch | 16.7 | 20.0 | 20.3 | 12.0 | 6.3 | 1.3 | 76.7 | 1.46 | 1.10 | 1.11 | 3.36 | 1.21 | 2.72 | 1.21 |
| | EvoEngineer-Solution (EoH) | 15.3 | 24.7 | 20.3 | 12.3 | 6.7 | 1.3 | 80.7 | 1.41 | 1.10 | 1.09 | 5.39 | 1.13 | 10.16 | 1.13 |
| | EvoEngineer-Free | 16.3 | 23.0 | 19.3 | 13.0 | 6.0 | 3.0 | 80.7 | 1.64 | 1.15 | 1.55 | 2.97 | 1.73 | 1.17 | 1.47 |
| | EvoEngineer-Insight | 15.7 | 19.3 | 19.3 | 13.0 | 6.7 | 3.0 | 77.0 | 1.46 | 1.23 | 1.55 | 3.48 | 2.16 | 9.13 | 1.59 |
| | EvoEngineer-Full | 16.3 | 21.7 | 19.3 | 12.0 | 6.3 | 2.7 | 78.3 | 1.43 | 1.07 | 1.08 | 11.96 | 1.74 | 16.61 | 1.20 |
| **Claude-Sonnet-4** | AI CUDA Engineer | 16.7 | 22.0 | 19.7 | 13.3 | 7.0 | 3.3 | 82.0 | 3.18 | 1.21 | 1.10 | 4.39 | 1.44 | 10.62 | 1.31 |
| | FunSearch | 16.3 | 21.3 | 20.3 | 13.3 | 5.3 | 3.3 | 80.0 | 2.40 | 1.15 | 1.11 | 10.70 | 1.46 | 13.41 | 1.31 |
| | EvoEngineer-Solution (EoH) | 16.7 | 19.7 | 19.7 | 14.0 | 6.7 | 3.3 | 80.0 | 3.38 | 1.32 | 1.26 | 2.61 | 2.03 | 16.87 | 1.59 |
| | EvoEngineer-Free | 16.3 | 22.3 | 21.0 | 13.7 | 6.7 | 3.0 | 83.0 | 2.26 | 1.54 | 9.23 | 12.78 | 11.21 | 18.35 | 2.72 |
| | EvoEngineer-Insight | 15.0 | 20.7 | 14.7 | 13.7 | 7.0 | 3.3 | 74.3 | 2.37 | 1.31 | 1.21 | 6.34 | 2.25 | 9.17 | 1.47 |
| | EvoEngineer-Full | 15.7 | 23.7 | 17.3 | 10.3 | 6.7 | 3.3 | 77.0 | 1.54 | 1.07 | 1.06 | 1.60 | 1.48 | 14.53 | 1.14 |

| Validity Analysis | | Compilation Success (Pass@1) | | | | | | | Functional Correctness (Pass@1) | | | | | | |
| --- | --- | --- | --- | --- | --- | --- | --- | --- | --- | --- | --- | --- | --- | --- | --- |
| | | 1 | 2 | 3 | 4 | 5 | 6 | Overall | 1 | 2 | 3 | 4 | 5 | 6 | Overall |
| **GPT-4.1** | AI CUDA Engineer | 88.3 | 85.7 | 82.7 | 82.1 | 86.8 | 62.2 | 84.0 | 59.7 | 52.7 | 70.2 | 56.9 | 73.1 | 29.6 | 59.4 |
| | FunSearch | 83.7 | 76.0 | 69.0 | 69.3 | 75.2 | 50.2 | 73.8 | 54.0 | 51.0 | 61.4 | 47.5 | 63.9 | 22.8 | 53.2 |
| | EvoEngineer-Solution (EoH) | 87.0 | 68.7 | 77.2 | 68.7 | 82.1 | 58.4 | 74.7 | 61.9 | 40.8 | 67.5 | 50.2 | 67.1 | 24.7 | 53.7 |
| | EvoEngineer-Free | 83.6 | 77.8 | 66.9 | 77.2 | 65.2 | 49.8 | 74.1 | 50.4 | 50.7 | 58.6 | 54.6 | 54.4 | 23.9 | 52.2 |
| | EvoEngineer-Insight | 83.7 | 80.9 | 83.5 | 81.3 | 90.2 | 66.2 | 66.0 | 56.0 | 47.8 | 74.4 | 65.1 | 73.2 | 39.2 | 60.0 |
| | EvoEngineer-Full | 87.9 | 86.0 | 87.4 | 90.2 | 92.9 | 77.3 | 87.5 | 66.6 | 59.5 | 79.8 | 75.6 | 83.1 | 55.8 | 69.8 |
| **DeepSeekV3.1** | AI CUDA Engineer | 93.5 | 80.9 | 76.9 | 73.6 | 89.0 | 45.0 | 80.3 | 64.2 | 47.5 | 62.3 | 43.2 | 62.0 | 35.0 | 54.0 |
| | FunSearch | 92.0 | 85.6 | 69.0 | 72.6 | 67.1 | 44.6 | 78.4 | 70.3 | 65.4 | 61.2 | 45.0 | 56.1 | 40.0 | 60.8 |
| | EvoEngineer-Solution (EoH) | 88.0 | 83.3 | 86.8 | 74.3 | 91.2 | 41.6 | 82.2 | 54.4 | 49.2 | 70.3 | 38.1 | 66.2 | 23.7 | 53.2 |
| | EvoEngineer-Free | 84.7 | 68.0 | 59.0 | 66.4 | 60.0 | 42.8 | 67.2 | 51.8 | 52.1 | 48.6 | 41.6 | 49.5 | 37.2 | 48.6 |
| | EvoEngineer-Insight | 96.7 | 83.5 | 90.7 | 83.9 | 91.9 | 75.6 | 88.1 | 74.2 | 50.4 | 74.6 | 58.2 | 66.0 | 54.8 | 63.4 |
| | EvoEngineer-Full | 97.8 | 86.9 | 92.7 | 87.8 | 93.3 | 73.6 | 90.4 | 71.1 | 61.6 | 78.7 | 61.8 | 74.4 | 55.5 | 68.1 |
| **Claude-Sonnet-4** | AI CUDA Engineer | 91.2 | 82.9 | 77.7 | 68.5 | 84.2 | 74.8 | 80.7 | 59.6 | 47.8 | 65.9 | 46.3 | 67.4 | 44.3 | 55.4 |
| | FunSearch | 94.6 | 82.2 | 86.1 | 76.4 | 70.2 | 76.5 | 83.6 | 65.6 | 49.2 | 77.9 | 54.1 | 53.3 | 55.2 | 60.6 |
| | EvoEngineer-Solution (EoH) | 87.8 | 71.2 | 83.6 | 64.8 | 71.9 | 62.2 | 75.7 | 48.7 | 31.4 | 67.0 | 40.3 | 51.7 | 26.3 | 45.5 |
| | EvoEngineer-Free | 84.2 | 85.7 | 76.5 | 75.9 | 82.2 | 73.9 | 80.9 | 55.6 | 51.6 | 66.5 | 53.1 | 62.0 | 48.7 | 56.8 |
| | EvoEngineer-Insight | 94.7 | 87.8 | 57.4 | 81.1 | 97.1 | 84.2 | 81.6 | 59.9 | 54.4 | 45.1 | 62.5 | 88.5 | 71.5 | 58.0 |
| | EvoEngineer-Full | 96.3 | 91.1 | 69.9 | 50.0 | 93.7 | 84.9 | 80.1 | 60.5 | 64.5 | 60.1 | 38.4 | 83.9 | 64.6 | 59.6 |

For code validity assessment, we record the ratio of valid kernels. The validity check involves compilation and correctness verification against the reference Python implementation. The Pass@1 metric is reported, indicating the proportion of trials that yield valid kernels.

## 5.2 OVERALL RESULTS

In a comprehensive evaluation across all 91 real-world CUDA kernels, EvoEngineer demonstrates superior performance,outperforming all competing methods in both speedup optimization and code validity. The overall results are summarized in Table 4.

**Speedup Performance.** EvoEngineer achieves substantial improvements in kernel optimization performance. The **EvoEngineer-Free** configuration paired with Claude-Sonnet-4 achieves a median speedup of **2.72×**, outperforming all competing approaches. This superior performance is consistent across all evaluated LLMs, demonstrating the framework's robustness and applicability.

**Code Validity Performance.** EvoEngineer maintains high code validity while achieving speedup improvements. The **EvoEngineer-Full** configuration attains a code validity rate of **69.8%**, significantly exceeding all competing methods. This robustness is consistently validated across different LLMs. Notably, with Claude-Sonnet-4, where all methods achieve high validity rates, EvoEngineer attains the second-highest performance, further demonstrating the framework's stability.

**Cross-Model Ability.** Analysis across different LLMs reveals that models exhibit varying capabilities for different kernel categories. GPT-4.1 performs poorly on category 4 kernels, whereas DeepSeek-V3.1 and Claude-Sonnet-4 excel; conversely, GPT-4.1 significantly outperforms others on category 5 kernels. Therefore, leveraging model-specific strengths for different kernel categories represents a promising direction for future work.

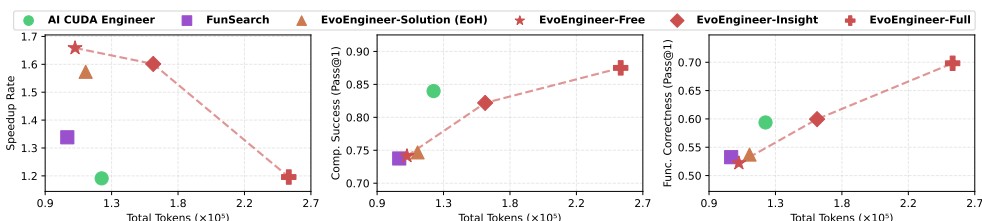

Figure 4: Token usage analysis for different methods using GPT-4.1.

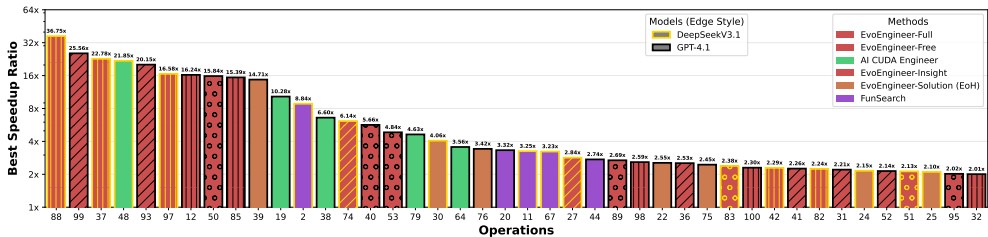

Figure 5: Speedup over **2×** compared to PyTorch kernels.

## 5.3 TOKEN USAGE ANALYSIS

Beyond its ability to balance performance and correctness in a principled manner, EvoEngineer enables configurable trade-offs between token usage and performance, as described in Section 4.1.1. Figure 4 presents the speedup and code validity plotted against token usage for different methods evaluated with GPT-4.1. Additional results using other LLMs are provided in Appendix A.5.

**EvoEngineer-Free** can utilize minimal tokens to explore diverse optimization strategies, achieving high performance while maintaining code validity. Alternatively, **EvoEngineer-Full** can invest additional tokens to further enhance code validity while maintaining performance competitive with the AI CUDA Engineer method. **EvoEngineer-Insight** strikes a balance between token usage, performance, and validity, demonstrating the framework's flexibility.

## 5.4 COMPARISON WITH PYTORCH KERNELS

While improving upon an initial LLM-generated kernel measures an optimization algorithm's search capability, comparing final kernels against highly-optimized, industry-standard libraries provides a more practical evaluation. We benchmark the final kernels produced by each method against PyTorch's default implementations. The complete results are provided in Appendix A.6.

Figure 5 presents operations achieving over **2×** speedup compared to PyTorch kernels. For each operation, we report the maximum speedup achieved across all methods and LLMs.

Of the 42 operations, EvoEngineer delivers the highest speedup on **25** (**59.5%**) operations, demonstrating its superiority in discovering high-performance kernels. Among EvoEngineer variants, **EvoEngineer-Full** achieves the best performance on 10 operations, demonstrating that our investigation of token usage yields practical performance gains. Notably, Claude-Sonnet-4 enables no method to achieve the highest speedup on any operation, highlighting the importance of model selection for performance optimization.

## 6 CONCLUSION

We addressed the fragmented landscape of automated CUDA kernel optimization by introducing EvoEngineer, the first systematic framework that decomposes LLM-based code evolution into orthogonal components: traverse techniques and population management. This decomposition enables principled strategy selection to balance performance and correctness. Our validation across 91 real-world CUDA kernels demonstrates EvoEngineer's effectiveness, achieving **2.72×** median speedup with **69.8%** code validity, substantially outperforming existing methods. Moreover, EvoEngineer attains the highest speedup on **59.5%** of operations with over **2×** acceleration for PyTorch kernels. Our work establishes a robust foundation for future research in automated kernel optimization.

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

# A APPENDIX

## A.1 LLM-BASED KERNEL ENGINEERING METHODS

**Kernel Code Evaluation.** A foundational contribution to this field is KernelBench Ouyang et al. (2025), a benchmark designed specifically to evaluate the ability of LLMs to generate efficient GPU kernels. It offers a standardized suite of deep learning tasks, reference Python implementations, and an evaluation framework for assessing both the correctness and performance of generated kernels.

Using a fixed prompt template, it instructs LLMs to produce CUDA kernels directly, thereby evaluating their capacity to write efficient code without requiring further optimization. By establishing a common basis for comparison, KernelBench has catalyzed research in automated kernel generation and serves as the primary evaluation standard for subsequent methods.

**Kernel Code Generation.** Several methods have been proposed to iteratively generate and refine kernel code leveraging the metrics and benchmarks defined by KernelBench. Sakana AI's AI CUDA Engineer employs an agent-based system that decomposes the optimization process into distinct stages, using LLM inference within feedback loops to systematically improve kernel performance Lange et al. (2025). The system operates through four main stages: Convert (transforming high-level descriptions to initial implementations), Translate (converting between different programming paradigms), Optimize (applying performance improvements), and Compose (combining optimization strategies).

Similarly, NVIDIA developed a closed-loop workflow that pairs the DeepSeek-R1 model with a validator, iteratively refining prompts to enhance the correctness and efficiency of the generated code Chen et al. (2025a). This approach emphasizes the importance of model-validator interaction in achieving consistent performance improvements.

Other notable contributions include CUDA-LLM Chen et al. (2025b) and CUDA-L1 Li et al. (2025) for NVIDIA GPUs, which focus on domain-specific optimizations and architectural-aware code generation. For AMD GPUs, GPU Kernel Scientist Andrews & Witteveen (2025) and GEAK Wang et al. (2025) have been developed to address the unique characteristics of AMD's GPU architectures and programming models.

**General-Purpose Code Generation.** In general-purpose code generation domain, AlphaCode Li et al. (2022) pioneered the use of LLMs for CUDA kernel generation, demonstrating the potential of LLMs to handle complex programming tasks.

## A.2 EXPERIMENTAL SETUP DETAILS

**Hardware Configuration** All experiments were conducted on a dedicated computing system with the following specifications:

- **CPU**: AMD EPYC 7542 32-Core Processor (2.9/3.4 GHz base/boost)
- **GPU**: NVIDIA RTX 4090 (16,384 CUDA cores, 24GB GDDR6X, 1008 GB/s)
- **System Memory**: 64GB DDR4-3200 ECC
- **Storage**: 50GB NVMe SSD for fast data access

The CPU performance directly impacts kernel launch overhead and memory transfer operations, while the GPU specifications determine the computational resources and memory bandwidth available for kernel execution. We selected the RTX 4090 due to widespread adoption in both research and industrial settings, ensuring practical relevance of our findings.

**Software Environment** The experimental platform is under the following software configuration:

- **Operating System**: Ubuntu 22.04 LTS
- **Python**: Version 3.11.11
- **PyTorch**: Version 2.4.0 with CUDA support (`torch==2.4.0+cu124`)
- **CUDA**: Version 12.4.1

These specific library versions are critical for reproducibility, as they define the available CUDA APIs, compiler optimizations, and runtime behaviors that influence kernel performance.

**Reproducibility Guidelines:**

1. Use identical GPU architecture (RTX 4090 or equivalent architecture)
2. Install the exact software versions listed above

Table 5: Kernel Classification by Computational Complexity.

| Category | Description | Count (%) |
|---|---|---|
| Matrix Multiplication | $\mathcal{O}(n^3)$ or higher complexity, highly parallel | 18 (19.8%) |
| Convolution | Multi-dimensional sliding window, complex memory access | 28 (30.8%) |
| Activation & Pooling | Element-wise, highly parallel | 21 (23.1%) |
| Normalization & Reduction | Statistical computation, dimension reduction | 15 (16.5%) |
| Loss Functions | Training optimization objectives | 7 (7.7%) |
| Cumulative Operations | Sequence dependent, hard to parallelize | 5 (5.5%) |
| **Total** | | **91 (100%)** |

Table 6: Model Abbreviations and Token Pricing.

| Abbreviation | Full Model Name | Input ($/M tokens) | Output ($/M tokens) |
|---|---|---|---|
| GPT-4.1 | `gpt-4.1-2025-04-14` | 2.00 | 8.00 |
| DeepSeekV3.1 | `deepseek-v3-1-250821` | 0.56 | 1.68 |
| Claude-Sonnet-4 | `claude-sonnet-4-20250514` | 3.00 | 15.00 |

3. Set consistent CUDA compilation flags: `-O3 -arch=sm_89 --use_fast_math`

4. Run each kernel multiple times (*e.g.,* 100 iterations) and report average execution time

Complete environment setup scripts are available in our public repository.

## A.3 DATASET DETAILS

Matrix multiplication operations (19.8%) have the highest computational complexity with $\mathcal{O}(n^3)$ or higher complexity. Convolution operations (30.8%) involve complex memory access patterns. Activation and pooling operations (23.1%) are element-wise with low complexity. Normalization and reduction operations (16.5%) require statistical computations. Loss functions (7.7%) are used for training objectives, and cumulative operations (5.5%) have sequence dependencies that make parallelization challenging. The operations are categorized by computational complexity and functionality, as shown in Table 5.

## A.4 PARAMETER SETTINGS FOR METHODS

In this section, we provide additional details on the settings and configurations used for the various methods evaluated in our experiments. We standardized the computational budget across all methods, setting the maximum number of optimization trials for each kernel to 45.

For AI CUDA Engineer, we followed the experimental setup from its original paper (Lange et al., 2025), which uses a maximum of 45 trials with diversified proposals by four distinct LLMs across 10 generations plus 5 RAG-based proposals ($4 \times 10 + 5 = 45$). We maintain the same configuration in our experiments, differing only in the LLM model used.

For FunSearch, we set the number of islands to 5 and continue sampling until reaching the maximum number of trials. For EoH, we set the population size to 4 and run the evolution for 10 generations. The initialized population consumes 5 trials, then in each generation the **E1**, **E2**, **M1**, and **M2** operators (Liu et al., 2024) are used to produce 4 offsprings, after which the top 4 individuals are selected to form the next generation. This process is repeated for 10 generations, with each generation producing 4 offspring, resulting in $4 \times 10 + 5 = 45$ trials.

For different variants of EvoEngineer, **EvoEngineer-Free** and **EvoEngineer-Insight** allow a maximum of 45 trials, while **EvoEngineer-Full** uses the same settings as EoH.

**Model Details.** We incorporate **GPT-4.1** from OpenAI, **DeepSeek-V3.1** from DeepSeek, and **Claude–Sonnet-4** from Anthropic. The detailed model versions and pricing information about them are provided in Table 6.

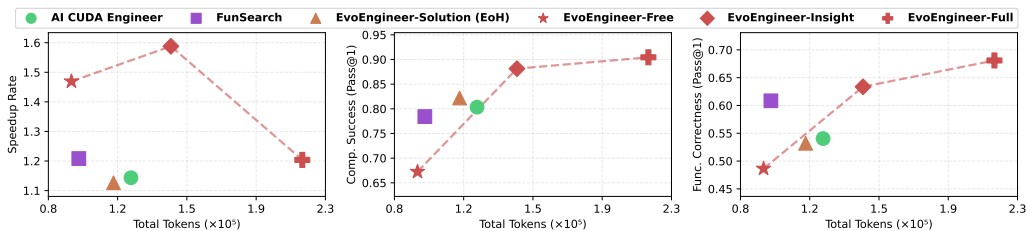

Figure 6: Token usage analysis for different methods using DeepSeek-V3.1.

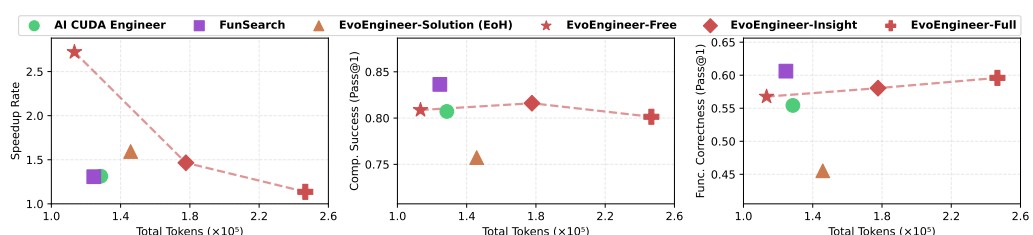

Figure 7: Token usage analysis for different methods using Claude-Sonnet-4.

## A.5    MORE ON TOKEN USAGE ANALYSIS

To further investigate the token usage patterns, we analyze the tokens consumed across different methods and their respective optimization stages. This analysis helps identify potential bottlenecks and areas for improvement in LLM interactions.

## A.6    COMPARISON TO PYTORCH KERNELS

While improving upon an initial LLM-generated kernel is a key measure of an optimization algorithm's search capability, a more practical assessment involves comparing the final kernels to highly-optimized, industry-standard libraries. In this section, we benchmark the final kernels produced by each method against the default implementations in PyTorch. The results are presented in Table 7 and visualized in Figure 8.

**Speedup Distribution Analysis.** Table 7 reveals significant performance variations across different methods and LLM combinations. EvoEngineer variants demonstrate competitive performance in achieving high-performance optimizations, with different configurations showing distinct strengths across speedup ranges.

**Cross-Model Performance Patterns.** The analysis reveals distinct model-specific optimization capabilities. GPT-4.1 demonstrates the most balanced performance, with EvoEngineer-Free achieving the lowest failure rate (26 kernels $<1.0\times$) and competitive high-performance results (3 kernels each in $5.0$-$10.0\times$ and $>10.0\times$ ranges). DeepSeekV3.1 shows more conservative patterns, yet EvoEngineer-Full achieves 4 kernels in the $>10.0\times$ category, demonstrating that comprehensive information integration can compensate for model limitations.

**Method-Specific Insights.** EvoEngineer variants exhibit complementary optimization profiles: EvoEngineer-Free consistently delivers strong exploration capabilities with minimal resource usage, EvoEngineer-Insight balances performance and validity (achieving the best $2.0$-$5.0\times$ results with GPT-4.1), while EvoEngineer-Full maximizes extreme performance potential. This systematic variation validates our framework's ability to provide tailored optimization strategies for different performance-correctness trade-offs.

Table 7: **Distribution of speedup ranges across methods and LLMs.** Results represent the maximum performance achieved across three independent runs. Best results are highlighted in **bold** with gray background ▨; second-best results are underlined.

| Model | Method | Count for Speedup Range | | | | |
|---|---|---|---|---|---|---|
| | | $<1.0$ | $1.0\sim2.0$ | $2.0\sim5.0$ | $5.0\sim10.0$ | $>10.0$ |
| **Claude-Sonnet-4** | AI CUDA Engineer | 24 | **43** | 17 | 1 | 4 |
| | FunSearch | 33 | 38 | 14 | 2 | 2 |
| | EvoEngineer-Solution (EoH) | 33 | 31 | **19** | 0 | **5** |
| | **EvoEngineer-Free** | 30 | 35 | 16 | **3** | 4 |
| | **EvoEngineer-Insight** | **34** | 32 | 15 | 3 | 1 |
| | **EvoEngineer-Full** | 30 | 34 | 16 | 0 | 3 |
| **DeepSeekV3.1** | AI CUDA Engineer | 35 | 35 | 13 | 2 | 2 |
| | FunSearch | 36 | 35 | 12 | 2 | 0 |
| | EvoEngineer-Solution (EoH) | 35 | 35 | 15 | 1 | 1 |
| | **EvoEngineer-Free** | 35 | 33 | 14 | **2** | 2 |
| | **EvoEngineer-Insight** | 35 | **35** | **15** | 1 | 1 |
| | **EvoEngineer-Full** | **37** | 35 | 11 | 0 | **4** |
| **GPT-4.1** | AI CUDA Engineer | 31 | **42** | 11 | **4** | 2 |
| | FunSearch | 29 | 41 | 14 | 2 | 2 |
| | EvoEngineer-Solution (EoH) | **32** | 37 | 15 | 1 | 3 |
| | **EvoEngineer-Free** | 26 | 40 | 14 | 3 | **3** |
| | **EvoEngineer-Insight** | 28 | 37 | **15** | 2 | 3 |
| | **EvoEngineer-Full** | 31 | 37 | 14 | 3 | 2 |

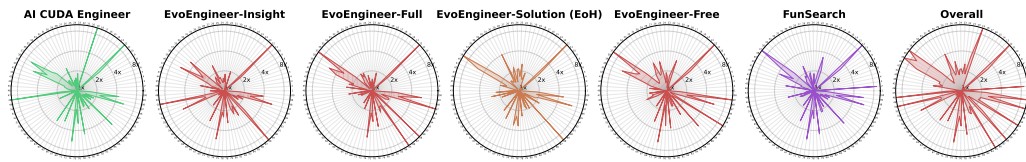

Figure 8: Speedup performance distributions across optimization methods. Each method displays maximum speedup achieved across three independent runs and different LLMs. The overall category shows the best performance achieved across all methods.

## A.7 DISCUSSION

In this section, we discuss acknowledge the limitations of this study and outline promising directions for future research.

### A.7.1 LIMITATIONS AND THREATS TO VALIDITY

Despite the promising results, we acknowledge several limitations that bound our conclusions and offer avenues for future work.

- **Hardware Specificity:** Our experiments were conducted on a single high-end NVIDIA GPU (RTX 4090). High-performance kernels are notoriously sensitive to the underlying architecture. The generalizability of our findings to other platforms, such as AMD GPUs or different NVIDIA architectures (e.g., Hopper), has not yet been empirically validated.

- **LLM Dependency:** The performance of EvoEngineer is inherently coupled with the capabilities of the underlying LLMs. The quality of generated code variations is contingent on the model's understanding of CUDA and parallel programming concepts. While more capable future models will likely enhance performance, this creates a dependency on specific, often proprietary, models.

- **Stochasticity of Performance Measurement:** Kernel execution times exhibit natural variance due to factors like system load, cache state, and GPU clock frequency fluctuations. While we mitigated this by taking the median of multiple runs, this stochasticity can still

influence the selection process in the evolutionary algorithm, potentially causing a genuinely superior kernel to be discarded based on a single noisy measurement.

### A.7.2 FUTURE RESEARCH DIRECTIONS

Building on this work, several exciting research avenues emerge:

- **Cross-Platform Generalization:** A crucial next step is to extend our modular framework to support a wider range of hardware, including AMD GPUs (via ROCm/HIP) and other accelerators. This would involve abstracting the hardware-specific aspects of the evaluator and developing new prompt strategies tailored to different programming models.
- **Advanced Evolutionary Techniques:** The current implementation of EvoEngineer uses a relatively standard evolutionary algorithm. There is considerable scope to incorporate more sophisticated techniques, such as multi-objective optimization (e.g., simultaneously optimizing for latency, power consumption, and register usage) or co-evolving kernels with their compilation parameters.
- **Enhanced LLM Interaction and Agency:** The interaction with the LLM could be made more dynamic. Instead of simply generating code from a static prompt, a future system could engage in a dialogue with the LLM, prompting it to explain its reasoning or critique its own suggestions. This would move the system towards a more powerful, agent-like paradigm for automated performance engineering.

### A.8 REPLICATION OF AI CUDA ENGINEER

In this work, we replicated Sakana AI's AI CUDA Engineer. We adhered strictly to the methodology outlined in the original paper Lange et al. (2025), ensuring a faithful replication.

### A.8.1 TECHNICAL DETAILS

**Workflow.** The workflow of AI CUDA Engineer, as outlined in the paper, consists of four stages: (1) *Convert*, (2) *Translate*, (3) *Optimize*, and (4) *Compose*. Since the prompts for the first three stages are provided in the original study, we directly adopt them in our replication, supplementing additional details where necessary. The following sections elaborate on the replication process for each step.

*Convert.* We employ the prompt from the original paper to convert the code, introducing two modifications: (1) we incorporate example content using code from the dataset released by Sakana AI AI (2025b), and (2) we impose a retry limit of 10 to prevent infinite loops. If the LLM fails to convert the code after 10 attempts, the process terminates, and we classify the optimization for this instance as a failure.

*Translate.* We follow the same approach as in the *Convert* step, including the addition of example code and a retry limit of 10. However, unlike the conversion stage, translation failures do not halt the process. Instead, we proceed with the translated code, even if it contains errors.

*Optimize.* For this step, we retain the original prompting strategy, which includes: *1)* providing five correct kernels, *2)* employing ensemble prompting, *3)* incorporating high-level information, and *4)* supplying profiling information. Although crossover prompting is mentioned for this step, the original paper does not provide specific prompts or implementation details. Consequently, we omit crossover prompting in our replication, as it remains unclear whether the original results were obtained with or without it.

*Compose.* This stage, also referred to as RAG-based *Optimize*, lacks a dedicated prompt in the original paper. Therefore, our replication involves a two-step process: *1)* we first select top 5 kernels from other kernels in the dataset using the embedding and search algorithm described in the original study, and *2)* we then adapt the prompt from the *Optimize* stage and append the selected kernel code to the prompt.

**Evaluation Method.** The evaluation process in the original paper is not explicitly detailed, but we can infer it from the evaluation scripts provided in their kernel archive AI (2025a). We follow the

Table 8: Overall Performance of Engineer. \*: The results are evaluated using closed-source script. NOT COMPARABLE.

|  | Reported\* | Tested | Ours |
| --- | --- | --- | --- |
| Median Speedup (all) | 1.13 | 0.82 | 1.10 |
| Median Speedup (success) | 1.52 | 1.66 | 2.94 |
| Sucessful Tasks ($>$ 1x speedup) | 63 | 22 | 49 |

same evaluation process as the original paper, which includes a correctness test of 5 random inputs and a performance test using the `torch.utils.benchmark.Timer().timeit` library.

### A.8.2 EXPERIMENTAL SETUP.

**Evaluation Dataset.** The framework was evaluated on the KernelBench dataset Ouyang et al. (2025), the latest version of which includes four tiers of PyTorch operators: level 1 (single-kernel operators ), level 2 (simple fusion patterns ), level 3 (full model architectures), and level 4 (Level Hugging Face). Our evaluation focused on the first three levels, aligning with the original study. Due to time constraints during replication, however, we validated our approach using only Level 1 kernels. Results for Levels 2 and 3 will be released in a subsequent publication.

**Environment.** All experiments were performed on an NVIDIA H100 GPU for CUDA kernel execution, using CUDA version 12.4.1, Python 3.11, and PyTorch 2.4.0. The choice of GPU and CUDA version is critical, as it directly impacts the underlying libraries utilized during kernel optimization. While the original paper does not specify the CUDA version, we selected version 12.4.1 due to its widespread adoption.

**Evaluation Method.** Since the evaluation process in the original paper is not explicitly detailed, we choose to follow the open-source evaluation process outlined in the KernelBench paper Ouyang et al. (2025) for presenting the results.

**Parameter Setting.** We follow the parameter settings outlined in the original paper, with the following additional details:

- **Retry limit**: We set a retry limit of 10 for the *Convert* and *Translate* stages to prevent infinite loops.

- **RAG trial**: We set a limit of 5 for the *Compose* stage, which means that the LLM will only propose 5 kernels during this stage.

### A.8.3 RESULTS

**Overall End-to-end Performance.** The overall performance of AI CUDA Engineer is summarized in Table 8. The first column presents initial results affected by reward hacking, included solely for reference. These results were obtained using closed-source scripts, making them incomparable to the other two columns. The second column reports performance on the dataset released by Sakana AI AI (2025b), while the third column displays the results of our replication effort to reproduce these findings.

When evaluating the released dataset directly, the speedup decreased from 1.13x to 0.82x, and the number of successful tasks dropped from 63 to 22. This suggests that kernels identified using the reward-hacked script underperform relative to expectations. Nevertheless, the framework remains effective, as evidenced by the number of successful tasks (native 22, compile 60) and the median speedup (native 0.82, compile 1.38).

Our replication, with the reward hacking issue resolved, confirms that AI CUDA Engineer effectively optimizes kernels, achieving median speedups of 1.10x (native) and 1.19x (compile). Furthermore, the speedup for successful tasks improved significantly, reaching 2.94x (native) and 5.71x (compile).

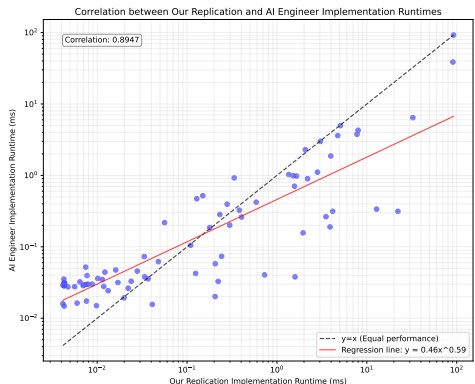

Figure 9: Correlation of Performance between our implementation with Sakana's paper.

**Validation of the Replication.** From an overall performance perspective, our replication achieves results comparable to those reported in the original paper. To further validate our replication, we examine the correlation in performance between our implementation and the original work. Specifically, we assess whether our kernel optimization capability aligns with that of the original study.

The results, presented in Figure 9, plot the speedup of the released dataset (x-axis) against the speedup achieved by our implementation (y-axis). As shown in the figure, our implementation exhibits a strong correlation with the original paper, with a coefficient of 0.9, confirming that our approach performs consistently with the original work.

## A.9 USAGE OF LARGE LANGUAGE MODELS

This work involves Large Language Models (LLMs) in two distinct capacities, which we disclose in accordance with academic transparency standards.

**Research Methodology.** LLMs constitute a fundamental component of our research methodology. We employ three state-of-the-art language models as core components of our EvoEngineer framework: GPT-4.1 (OpenAI), DeepSeek-V3.1, and Claude-Sonnet-4 (Anthropic). These models serve as the primary code generation and optimization engines within our evolutionary framework, generating and refining CUDA kernel implementations across 91 benchmark operations. All experimental results, performance evaluations, and comparative analyses presented in this paper directly reflect the capabilities and outputs of these LLM systems when integrated with our proposed optimization strategies.

**Manuscript Preparation.** During the preparation of this manuscript, we utilized Claude Code (Anthropic) as a writing assistant for various editorial and organizational tasks, including:

- LaTeX formatting and table/figure organization
- Proofreading and grammar correction

All core technical content, experimental design, algorithmic innovations, results interpretation, and scientific conclusions represent the original intellectual contributions of the authors. The use of AI assistance was limited to editorial support and did not influence the research direction, methodology, or findings. All claims and technical assertions have been independently verified by the research team.

