# OpenReview forum: "EvoEngineer: Mastering Automated CUDA Kernel Code Evolution with Large Language Models"
_ICLR.cc/2026/Conference — Submitted to ICLR 2026_

### Official Review · Reviewer_M7M3 · 2025-10-20

**Soundness:** 3
**Presentation:** 3
**Contribution:** 2
**Rating:** 2
**Confidence:** 3

**Summary:**

This paper proposes EvoEngineer, a systematic framework that utilizes large language models to automatically optimize CUDA kernel code. It classifies prompts to explore the impact of different prompt types on the results. The framework also supports the verification of syntactic correctness and performance testing, feeding the results back to the large language model for iteration. Finally, a population management component selects the high-quality solutions provided by the LLM.

**Strengths:**

1. The method proposed in the paper performs well and has been verified on multiple LLMs.

2. Significant effort was invested in replicating the key baseline, AI CUDA Engineer, which greatly enhances the credibility of the comparison results.

**Weaknesses:**

1.The experiments rely entirely on expensive large-scale models (GPT-4.1, DeepSeek-V3.1, Claude-Sonnet-4), without evaluating low-cost LLMs such as Qwen3-32B or Qwen3-Next (or even Qwen-7B), thus limiting insights into the scalability of the framework in budget-constrained settings.

2.The lack of detailed prompt examples limits the reader’s understanding of the proposed prompt engineering process.

3.The experimental evaluation, though extensive, has several notable limitations. All experiments were conducted on a single RTX 4090 GPU, raising concerns about cross-platform generalizability. Performance measurements suffer from inherent stochasticity without statistical significance testing. Moreover, only Level-1 kernels from KernelBench were evaluated, excluding more complex fusion and model-level tasks.

4.The paper claims to propose an “LLM-driven evolutionary optimization” framework, but in practice it still relies on a static heuristic evolutionary algorithm without incorporating learning signals, reward modeling, or adaptive search.
As a result, the whole system behaves more like a stack of prompts with manually designed strategies rather than a genuinely intelligent evolutionary process.

**Questions:**

1.How does the proposed method perform on small-scale LLMs?

2.Although the tasks are not exactly the same, some common code optimization metrics[1] may be of reference.

[1]Shypula A, Madaan A, Zeng Y, et al. Learning performance-improving code edits[J]. arXiv preprint arXiv:2302.07867, 2023.

---

### Official Review · Reviewer_C1R6 · 2025-10-24

**Soundness:** 2
**Presentation:** 2
**Contribution:** 2
**Rating:** 2
**Confidence:** 4

**Summary:**

This paper introduces EvoEngineer, a framework for automated CUDA kernel code optimization leveraging large language models (LLMs). The authors formalize kernel optimization as a constrained code evolution problem, and propose a modular framework decomposing LLM-driven code evolution into two components: traverse techniques (including two sub-modules solution guiding and prompt engineering) and population management. The framework is instantiated into three concrete strategies based on the information the agent receive. The experiment are conducted on 91 real-world CUDA kernels from Kernel Bench, where EvoEngineer demonstrates best trade-offs between kernel speedup and code correctness compared to existing LLM-based and evolutionary code optimization methods.

**Strengths:**

-  By splitting LLM-based code optimization into "traverse techniques" and "population management," and further into a two-layer model, the method enables independent analysis of information flows and optimization effectiveness.
- The paper provides a constrained optimization formulation for CUDA kernel code evolution (Section 3.1, Equation 1).
- The study offers practical guidelines for evaluating code optimization methods by exploring the trade-offs among token/computational usage, speedup, and validity.

**Weaknesses:**

- While the two-layer "traverse technique" decomposition is justified (Section 4.1.1), the paper tends to overgeneralize the implications. Specifically, the paper asserts that all "operator-based" prior works conflate strategy with implementation, without a sufficiently granular breakdown or ablation study against competing operator definitions in, e.g., FunSearch.
- The mathematical formalization is restricted mainly to the high-level constrained optimization form (Equation 1), with little follow-through on more formal properties (e.g., landscape characterization, convergence of solution populations, or theoretical limitations of LLM-guided mutation/search in this discrete code space). The formal optimization statement is sound but does not translate into deeper theoretical results or predictive analysis of code search behavior.
- While Figures 4, 6, 7 present token-resource curves, the study's resource trade-off analysis is mostly empirical. There is little theoretical or algorithmic reasoning about the sensitivity (e.g. for different environments) or scaling (e.g., for different hardware settings).

**Questions:**

- The provided framework is described at an abstract level. What would be a concrete example?
- The evaluation of kernel correctness and speedup is performed primarily using median and count metrics. Could the authors provide further analysis of error and failure cases?
- Could the authors provide more insight into which operation categories benefit most and least from each EvoEngineer configuration, and explain why?
- During the search process, which optimization techniques (such as memory coalescing and occupancy optimization) are employed to accelerate the model? Which techniques are utilized most frequently?
- How sensitive is EvoEngineer to the selection and diversity of its "historical solutions" and "optimization insights" components? Have any ablation studies or diagnostic analyses been conducted to assess potential overfitting or instability arising from these information sources?
- How does the design of the prompt engineering layer affect code optimization performance?

---

### Official Review · Reviewer_35Jg · 2025-11-01

**Soundness:** 1
**Presentation:** 2
**Contribution:** 1
**Rating:** 0
**Confidence:** 4

**Summary:**

The paper proposes EvoEngineer, a system for automatic cuda code generation using LLMs and iterative refinement.

**Strengths:**

I believe the correctness results are not (or only marginally, re. reward hacking) impacted by my criticism of the evaluation below, and these show general improvements over previous methods.

**Weaknesses:**

unfortunately, the paper is hamstrung by the fact that I don't think any of the evaluations are meaningful.
while the paper itself is sparse in details, it generally defers to KernelBench, which (possibly unbeknownst to the authors) is critically flawed.
Just taking the first example from the linked repository:
> "org_py_code": "import torch\nimport torch.nn as nn\n\nclass Model(nn.Module):\n    \"\"\"\n    A model that computes Hinge Loss for binary classification tasks.\n\n    Parameters:\n        None\n    \"\"\"\n    def __init__(self):\n        super(Model, self).__init__()\n\n    def forward(self, predictions, targets):\n        return torch.mean(torch.clamp(1 - predictions * targets, min=0))\n\nbatch_size = 128\ninput_shape = (1,)\ndim = 1\n\ndef get_inputs():\n    return [torch.randn(batch_size, *input_shape), torch.randint(0, 2, (batch_size, 1)).float() * 2 - 1]\n\ndef get_init_inputs():\n    return []",

The input shape is `128 x 1`; there is no meaningful work to be done in this case, its just launch overhead.

In the past year, we've seen multiple claims of "high-performance" LLM-written cuda kernels followed by retractions because the evals were broken or the LLM was reward-hacking. Note that, e.g., the citation https://pub.sakana.ai/ai-cuda-engineer/leaderboard given in the paper 404s nowadays.

To  gain confidence that the results your seeing are real, you  need to
* ensure the inputs are large enough to justify launching a cuda kernel
* determine the speed-of-light of the kernels in question, i.e., calculate what the fastest possible execution of the kernel is based on  the GPUs memory bandwidth and flops; if the llm is faster than the speed-of-light, you know something is broken
* validate that you use proper baselines (e.g., torch.compile,  enabling tensor cores, ...); again,  speed-of-light can help to determine whether the baseline is unexpectedly weak
* carefully analyse individual cases with large speed-ups. Anything over 2x is highly suspicious, for matmul-like operations and other things that are used extensively in typical pytorch workloads, I'd consider anything over 10% suspicious, too.

**Questions:**

* what are the speed-ups on meaningful input shapes
* what are the numbers1-6 in Table 4

---

### Official Review · Reviewer_sPg4 · 2025-11-01

**Soundness:** 2
**Presentation:** 3
**Contribution:** 2
**Rating:** 4
**Confidence:** 3

**Summary:**

The paper proposes EvoEngineer, a systematic framework for automated CUDA kernel optimization using LLMs. This work formalizes kernel optimization as a constrained code evolution problem and decomposes LLM-based code evolution into two orthogonal components: (1) traverse techniques, with a two-layer design of “solution guiding” and “prompt engineering”, and (2) population management for solution maintenance. Based on the evaluation of 91 CUDA kernels, the authors claim up to 2.72× median speedup and 69.8% code validity, outperforming AI CUDA Engineer, FunSearch, and EoH.

**Strengths:**

1. The automation of CUDA kernel optimization is an important and timely topic, especially for AI system efficiency.
2. The paper is well organized and visually clear, with consistent formatting and informative tables.
3. The authors make an effort to discuss a taxonomy of existing LLM-based code optimization methods.
4. The experiment covers 91 CUDA kernels, demonstrating substantial engineering effort.

**Weaknesses:**

1. novelty: The claimed “systematic framework” introduces no genuine algorithmic innovation; the proposed two-layer design is largely conceptual and repackages standard evolutionary-search principles without introducing new operators or optimization rules.
2. The work lacks formalism and analytical grounding. Equation (1) remains a generic placeholder, with no derivation or justification for why the proposed decomposition improves search efficiency or correctness.
3. Experimental methodology is insufficiently detailed: key factors such as kernel size, difficulty level, GPU utilization, batch size, and warm-up policy are omitted, and no ablation is provided to isolate the contribution of each framework component. As a result, the empirical evidence does not convincingly support the claimed advantages of the approach.

**Questions:**

1. What is the key difference between your work and the current pipelines when they also do structured multi-stage optimization and correctness validation, especially when you are claiming as the first or unique?
2. your repo has only the data and results, making it hard to test the performance. Especially when the results are remarkble, such as up to 36X max speedup. How is the “two-layer traverse” implemented in code? Is there any measurable effect from separating prompt construction from solution guidance?
3. Why are no ablation studies provided to isolate the effects of (a) traverse technique design and (b) population strategy?

---

### Meta-Review · Area_Chair_DWfm · 2026-01-06

**Summary:**

Reviewers have little to no support for this paper and the authors did not rebutt.

**Reviewer Concerns:**

None was addressed.

**Reviewer Scores:**

All rejects.

---

### Decision · Program_Chairs · 2026-01-26

Reject